# Viral mapping in COVID-19 deceased in the Augsburg autopsy series of the first wave: A multiorgan and multimethodological approach

Klaus Hirschbühl[1]*, Sebastian Dintner[2], Martin Beer[3], Claudia Wylezich[3], Jürgen Schlegel[4], Claire Delbridge[4], Lukas Borcherding[2], Jirina Lippert[2], Stefan Schiele[5], Gernot Müller[5], Dimitra Moiraki[2], Oliver Spring[6], Michael Wittmann[1], Elisabeth Kling[7], Georg Braun[8], Thomas Kröncke[9], Rainer Claus[1], Bruno Märkl[2], Tina Schaller[2]

1 Hematology and Oncology, Medical Faculty, University of Augsburg, Augsburg, Germany, 2 General Pathology and Molecular Diagnostics, Medical Faculty, University of Augsburg, Augsburg, Germany, 3 Institute of Diagnostic Virology, Friedrich-Loeffler-Institute, Federal Research Institute for Animal Health, Greifswald-Insel Riems, Germany, 4 Department of Neuropathology, School of Medicine, Institute of Pathology, Technical University Munich, Munich, Germany, 5 Computational Statistics and Data Analysis, Institute of Mathematics, University of Augsburg, Augsburg, Germany, 6 Anesthesiology and Operative Intensive Care Medicine, Medical Faculty, University of Augsburg, Augsburg, Germany, 7 Microbiology, Medical Faculty, University of Augsburg, Augsburg, Germany, 8 Department of Gastroenterology, University Hospital Augsburg, Augsburg, Germany, 9 Diagnostic and Interventional Radiology, Medical Faculty, University of Augsburg, Augsburg, Germany

☯ These authors contributed equally to this work.
* klaus.hirschbuehl@uk-augsburg.de

**Data Availability Statement:** All relevant data are within the manuscript and its Supporting Information files.

## Abstract

### Background

COVID-19 is only partly understood, and the level of evidence available in terms of patho-physiology, epidemiology, therapy, and long-term outcome remains limited. During the early phase of the pandemic, it was necessary to effectively investigate all aspects of this new disease. Autopsy can be a valuable procedure to investigate the internal organs with special techniques to obtain information on the disease, especially the distribution and type of organ involvement.

### Methods

During the first wave of COVID-19 in Germany, autopsies of 19 deceased patients were performed. Besides gross examination, the organs were analyzed with standard histology and polymerase-chain-reaction for SARS-CoV-2. Polymerase chain reaction positive localizations were further analyzed with immunohistochemistry and RNA-in situ hybridization for SARS-CoV-2.

### Results

Eighteen of 19 patients were found to have died due to COVID-19. Clinically relevant histological changes were only observed in the lungs. Diffuse alveolar damage in considerably

**Funding:** Funding of Corona research projects by the Bavarian State Ministry of Science and Art https://www.stmwk.bayern.de/ The funders had no role in study design, data collection and analysis, decision to publish, or preparation of the manuscript.

**Competing interests:** The authors have declared that no competing interests exist.

different degrees was noted in 18 cases. Other organs, including the central nervous system, did not show specific micromorphological alterations. In terms of SARS-CoV-2 detection, the focus remains on the upper airways and lungs. This is true for both the number of positive samples and the viral load. A highly significant inverse correlation between the stage of diffuse alveolar damage and viral load was found on a case and a sample basis. Mediastinal lymph nodes and fat were also affected by the virus at high frequencies. By contrast, other organs rarely exhibited a viral infection. Moderate to strong correlations between the methods for detecting SARS-CoV-2 were observed for the lungs and for other organs.

## Conclusions

The lung is the most affected organ in gross examination, histology and polymerase chain reaction. SARS-CoV-2 detection in other organs did not reveal relevant or specific histological changes. Moreover, we did not find CNS involvement.

## Introduction

COVID-19, which emerged at the end of 2019, is only partly understood, and the level of evidence available in terms of pathophysiology, epidemiology, treatment, and long-term outcomes is limited. During the early phase of the pandemic, the clinical course of this novel infectious disease was found to differ from that of other viral infections such as influenza. COVID-19 presents more frequent with severe illness accompanied with lung injury which results more often in fatal course, and also SARS-CoV-2 seems to be more contagious and virulent than influenza [1–3]. Given the high numbers of critically ill patients, it became necessary to establish an effective approach to investigate all the aspects of this new disease. Autopsy rates declined dramatically over the last decades, but it can offer the opportunity to directly inspect all organs, identify the cause of death, and generate specimens for further investigations [4]. And this is true for COVID-19 as well, a new disease still not yet understood in detail and lack of sufficient therapeutic approach. Therefore, the aim was to analyze the macroscopic, microscopic changes and the spread of the virus in the deceased patients for the first step and to generate tissue and liquid samples for further investigations in terms of mechanisms of the disease. After overcoming initial restrictions regarding performing autopsies in cases of COVID-19, especially in Europe, several pathologists and forensic scientists of different departments started structured activities in this direction [5–8]. The main finding from autopsies is diffuse alveolar damage (DAD) [9, 10]. The lung changes occur independently of the performance of invasive ventilation [11]. In addition, reports of thromboembolic events at different levels—ranging from microthrombi to fulminating macroembolism with extensive organ infarction [12]. Varga et al. first proposed endothelialitis as a cause of this increased coagulation propensity [13]. Renal damage has also been described in detail [14]. However, the findings related to the central nervous system (CNS) have been contradictory and do not yet provide sufficient evidence for direct brain damage by SARS-Cov2, except for involvement in systemic coagulation.

Here, we present a series of 19 postmortem examinations of COVID-19 patients with complete gross examination in most cases and analysis of organs, including the CNS. We conducted a comprehensive approach for multimodal viral mapping within the organism applying different methods: immunohistochemistry (IHC), real-time quantitative polymerase

chain reaction (RT-qPCR), and RNA-in situ hybridization (RNA-ISH) (positive/plus and negative/minus sense). Additionally, we evaluated the correlation of DAD with viral load in the lungs and with clinical parameters.

## Methods and materials

### Cohort

From April 4 to May 13, 2020, 19 autopsies (15 full and 4 limited) were performed. 18 patients died at the University Medical Center Augsburg (UKA), representing 86% of all Covid-19 deceased at the UKA during the first wave of the pandemic. Only one patient was treated and died in a community hospital outside Augsburg. All cases were tested positive for SARS-CoV-2 with nasopharyngeal swabs during the clinical course.

According to national rules after the death of the patient and before autopsy, informed consent for the autopsy including research issues was obtained verbal by next of kin and documented in the patient´s record. Due to restricted consent in four cases, only limited autopsy with removal of parts of the lung, liver, kidney, spleen was performed. In these cases, also no CNS specimens could be collected. Clinical data (including medical history, comorbidities, medication, and treatment) of all persons who died were obtained from the electronic medical records. This study was approved by the internal review board of the UKA (BKF No. 2020–18) and the ethics committee of the University of Munich (Project number 20–426, COVID-19 registry of the UKA). Ten of 19 cases were already included in a previous [11].

### Autopsy

Depending on the consent, a conventional full autopsy or limited organ removal or biopsy was performed. Special attention was paid to the safety of personnel, as described in the previous study [15]. During inspection of the opened body cavities, particular attention was paid to effusion formation, hemorrhage, and signs of thromboembolic events. First, all removed organs (such as the lungs, heart, liver, spleen, kidneys, and brain) were inspected macroscopically, weighted, and measured. Next, they were immediately stored and fixed in buffered 10% formalin. For better fixation, the lungs were flushed with formalin by syringe injection. After at least 10 days of fixation, representative formalin-fixed and paraffin embedded samples (FFPE) were generated. During autopsy, swabs of different sides of the respiratory tract (the throat, trachea, and bronchus) were performed, and samples of CSF and pleural, peritoneal, or pericardial effusion (if present) were analyzed. In the cases of partial autopsies, in situ examinations of the lungs were performed and large biopsies of at least 8 cm in diameter were obtained.

### Histology and IHC (http://dx.doi.org/10.17504/protocols.io.bvvwn67e, S1 File)

From all FFPE blocks, hematoxylin and eosin (H&E)-stained slides for histological analysis were produced. Additionally, immunhistological staining for SARS-CoV-2 was performed from representative lung samples of the left and right lobes and from all samples with a SARS-CoV2-positive PCR test (see below). A monoclonal SARS N-specific antibody was used (4F3C4, 1:200, Institute of Virology, University of Leipzig, Leipzig, Germany) [16] and all steps were performed manually. The slides were deparaffinized and rehydrated in xylene and ethanol in decreasing concentrations. The endogenous peroxidase was deactivated with 3% hydrogen peroxide. For antigen retrieval, microwave treatment (700 W, 20 min) was used. Unspecific binding was blocked using goat serum. The primary antibody was incubated

overnight at 4˚C, the secondary antibody (goat antimouse biotin, 1:200, Vector laboratories, Peterborough, UK) was incubated for 30 min followed by incubation with the avidin–biotin complex for another 30 min. AEC-substrate chromogen was used to develop the staining reaction (Agilent-Dako, Waldbronn, Germany). The slides were counterstained with hematoxylin (Morphisto, Frankfurt, Germany). On-slide positive controls were selected from samples with positive virus detection and low CT values ($\leq 25$) and mounted on each slide.

All slides were evaluated by an experienced board-certified pathologist (TS). Based on the information obtained from this evaluation, particularly the extent of change seen, representative slides were selected and independently analyzed and classified by TS and another experienced board-certified pathologist (BM). Both of whom were blinded to the cases' clinical history. Discrepancies were resolved by mutual consensus. The histological analyses included the categorical assessment of several features in different organs. The categories were either dichotomous (not present/present) or four tiered (none, mild, intermediate, and strong). Immunohistochemical semiquantitative evaluation of SARS-CoV-2 was performed in a representative sample from each lobe of the lungs and in all samples of organs other than lungs with a positive result in the RT-qPCR. It was conducted by categorizing it into four groups: negative (no positive cells), low (a single or a few cells scattered in a low density), intermediate (strongly positive cells in a cluster of at least 5 mm diameter or weak/moderately positive in many cells throughout the sample), and high (many strongly positive cells throughout the sample).

## RT-qPCR (http://dx.doi.org/10.17504/protocols.io.bvvxn67n, S2 File)

RNA was extracted from 3 x 8-μm slices of FFPE tissue sections from all organs by using the Promega Maxwell® 16 MDx system and the Promega Maxwell 16 LEV RNA FFPE Purification Kit following the manufacturer's instructions (AS1260, Promega Corporation, Madison, WI, USA).

Quantitative real-time PCR for SARS-CoV-2 was performed on extracts with one-step multiplex RT-qPCR targeting the SARS-CoV-2 ORF1ab, N Protein, and S Protein using the Taq-Path COVID-19 CE-IVD RT-PCR Kit (A48067, Thermo Fisher, Pleasanton, TX, USA). Briefly, 5 μL of the extract was amplified in 25 μL of the reaction mixture containing 6.25 μL of TaqPath 1-Step Multiplex Master Mix, 1.25 μL of COVID-19 Real-Time PCR Assay Multiplex, and 12.5 of μL nuclease-free water, following the manufacturer's instructions. The RT-qPCR was run on the QuantStudio 5 Dx real-time PCR Instrument and data were analyzed and interpreted by QuantStudio™ design and analysis software (v.1.2x, Thermo Fisher, Carlsbad, CA, USA). As an extraction control, 10 μL MS2 phage control was added to the extraction and was used as a negative control in qPCR. The positive control was supplied by the TaqPath COVID-19 CE-IVD RT-PCR Kit. Results with two or more positive targets were described as valid.

The cycle-threshold (Ct) values were classified into seven categories (10–15; 16–20; 21–25; 26–30; 31–35; 36–40; negative).

## RNA-ISH (http://dx.doi.org/10.17504/protocols.io.bvvun66w, S3 File)

For the in-situ hybridization for SARS-CoV-2 on the FFPE materials of the lung specimens, two probes were used to detect infected cells and identify replicating viruses (positive/plus- and negative/minus-sense). Positive and negative controls were integrated to verify the correctness of the staining results (RNAscope Probe–V-nCoV2019-S and V-nCoV2019-S-sense, Positive Control Probe–Hs-UBC, Advanced Cell Diagnostics, Germany) RNAscope 2.5 HD Assay-RED kit (Advanced Cell Diagnostics, Germany) was used for realizing the hybridization

of the probes and its visualization according to the manufacture's protocols by manual technique on 5 μm thick FFPE sections.

RNA-ISH evaluation was performed on the same samples (lungs in all patients and for other organs, which were tested positive by RT-qPCR) as the immunohistochemical SARS-CoV-analysis. The classification scheme was also identical.

### Correlation of DAD with clinical parameters

As there are established or discussed clinical parameters and patient characteristics influencing the outcome of COVID-19, we aimed to correlate those parameters with the DAD stage. Therefore, we chose parameters which seems to be relevant from the clinical point of view and which were available for all 19 patients: Sex, age, smoking history, BMI, diabetes, numbers of comorbidities and length of ventilation.

### Statistical analysis

Correlations between Ct values and histological parameters, as well as the different RNA detection methods, were calculated using the Spearman rank order correlation. We calculated the Pearson correlation between DAD and several parameters of viral load in the lungs as well as clinical parameters. To detect the impact of clinical variables on DAD, we performed linear regression models and adjusted for potentially confounding variables through multiple linear regression models. All tests were two-sided, and $p < 0.05$ was set as significant. All analyses were conducted with free software environment for statistical computing and graphics R version 4.0.2 and Sigma plot 13.0 (Systat Software, San Jose, CA, USA).

## Results

### Cohort and clinical data

The median age of the 19 patients (5 women and 14 men) at the time of death was 73 years (range: 57–90 years). The patients had a median of four known preexisting comorbidities (range: 0–9), with cardiovascular disease being most frequent. Table 1 summarizes the patients' basic characteristics.

The most frequent initial symptoms included fever, cough, and dyspnea. During the clinical course, 15 patients were transferred to the intensive care unit. Nine of them required intubation and invasive ventilation, and five required noninvasive ventilation. In four cases, therapy was limited at the patient's request and intensive medical treatment was not initiated. A clinically diagnosed acute respiratory distress syndrome (ARDS) was reported in 16 patients. The median time from admission to death was 11 days (range: 1–36 days), and the median time between death and autopsy was 26 h (range: 6–53 h). Detailed information on clinical course and treatment is presented in Table 2.

### Autoptic macroscopical findings and histology

**Lungs.** Macroscopically, the lung tissue showed at least edema and slight redness, but most of them were markedly consolidated, hemorrhagic, and severely edematous and friable. These changes were accentuated in the lower lobes but were usually at least to some extent as well present in the other lobes. In one patient, extensive areas of sclerosis were observed on the parasagittal plains.

A total number of 680 slides from the lungs were screened, and 83 representative slides were selected by TS for the semiquantitative analyses that was conducted by TS and BM. Despite being obtained from autopsies, the histology was excellently preserved in most cases

**Table 1. Preexisting patients' basic characteristics.**

| Variable | No. |
|---|---|
| **Total number of patients** | 19 (100%) |
| Mean/Median Age (range) | 74/73 years (57–90) |
| Sex (male/female) | 14 (74%) / 5 (26%) |
| Smoker (yes/no) | 7 (37%) /12 (63%) |
| **Comorbidities** | |
| Median number (range) | 4 (0–9) |
| **Cardiovascular** | 13 (68%) |
| Atrial fibrillation | 11 (58%) |
| Coronary artery disease | 5 (26%) |
| Cardiomyopathy | 5 (26%) |
| Aortic valve stenosis | 1 (5%) |
| Hypertension | 13 (68%) |
| Arteriosclerosis | 9 (47%) |
| **Metabolic** | 11 (58%) |
| Diabetes | 6 (32%) |
| Obesity | 9 (47%) |
| Median body mass index (range) | 28.3 kg/m$^2$ (19.6–66.2) |
| **Chronic respiratory disease** | 5 (26%) |
| **Chronic renal disease** | 7 (37%) |
| **Hyperlipoproteinemia** | 4 (21%) |
| **Prevalent malignancies** | 3 (16%) |
| **Therapeutics** | |
| Angiotensin converting enzyme inhibitors | 9 (47%) |
| Angiotensin II receptor blockers | 2 (11%) |

CLL = chronic lymphatic leukemia, CMML = chronic myelomonocytic leukemia

with autolytic changes in the small minority only. Changes in DAD in considerably different degrees were observed in 72 of 83 available specimens from the lung lobs in all but one of the 19 cases (Fig 1). In this single case, only the right upper lope was affected by an acute DAD. The early stage (the highest stage in three cases), in addition to interstitial and intraalveolar edema, was characterized by hyaline membranes. Partly, a slight blood vessel congestion and swelling of the pneumocytes were appreciable. In the stage of organization (the highest stage in 15 cases), there was a broadening of alveolar septa and desquamation of pneumocytes. Macrophages were found mainly intraalveolar, whereas giant cells and lymphoplasmacytic inflammatory infiltrate of varying densities were found in the interstitial tissue and partly with perivascular accentuation. Furthermore, there was clear fibroblastic proliferation. In some sections, squamous, and osseous metaplasia were noted. In the late stage (so-called end-stage DAD; the highest stage in one case), the lung tissue shows scarring fibrosis and no longer an inflammatory infiltrate. Microthrombi were identified in 14 slides (17%) of 10 cases (53%). Areas with considerable vascular congestion and hemorrhage were observed in 30 (36%) slides in 11 (58%) cases.

**Heart, liver, kidney, spleen, lymph nodes, mediastinal fat.** 790 slides from organs other than the lungs and the CNS have been screened as described above. 134 of those have been selected for the semiquantitative evaluation.

On the basis of macroscopy and conventional light microscopy of H&E-stained slides, organs other than the lungs did not show any specific macroscopic or histological

**Table 2. Clinical course and treatment.**

| Variable | No. |
|---|---|
| **COVID-19 testing: nasopharyngeal swab** | 19 (100%) |
| **Symptoms on admission** | |
| Cough | 13 (68%) |
| Dyspnea | 11 (58%) |
| Fever | 9 (47%) |
| Worsening general condition | 9 (47%) |
| Diarrhea | 1 (5%) |
| Median time between onset of symptoms and admission* (range) | 5 days (0–21) |
| **Radiologic findings (CT = 4, X-Ray = 15)** | |
| Bilateral patchy shadowing | 16 (84%) |
| Ground-glass opacity | 2 (11%) |
| **Respiratory findings** | |
| Median Horowitz index* (range)) | 107 mmHg (50–150) |
| Acute respiratory distress syndrome | 16 (84%) |
| **Septic shock** | 7 (37%) |
| **Acute kidney injury** | 14 (74%) |
| **Congestive heart failure** | 13 (68%) |
| **Clinical diagnosed pneumonia** | 8 (42%) |
| **Thromboembolic events (DVT/PE)** | 2 (11%) |
| **Systemic therapy** | |
| Systemic glucocorticoids (hydrocortisone for septic shock) | 3 (16%) |
| Remdesivir | 0 (0%) |
| Chloroquine or Hydroxychloroquine | 8 (42%) |
| Reconvalescent plasma | 3 (16%) |
| Intravenous antibiotics | 17 (89%) |
| Median number of antibiotic substances (range) | 2 (0–8) |
| **Anticoagulation use** | |
| Deep venous thrombosis prophylaxis | 9 (47%) |
| Full dose anticoagulation | 10 (53%) |
| Vasopressors | 10 (53%) |
| **Ventilation** | |
| Oxygen only | 5 (26%) |
| Noninvasive | 5 (26%) |
| Invasive | 9 (47%) |
| **Renal replacement therapy** | 5 (26%) |

*Missing information in three cases

CT = computed tomography, DVT = deep vein thrombosis, PE = pulmonary embolism

abnormalities that would differ in type, extent, and frequency from autopsies of other deceased of the same age with similar previous diseases (for example, steatosis hepatitis, arteriosclerosis, vascular damage of the kidney). Frequently found changes were related to cardiovascular changes: cardiomyopathy, coronary heart disease (two cases with old myocardial infarction), and general arteriosclerosis. Hepatic changes included steatosis and fibrotic changes accompanied by mild portal inflammation. Relevant kidney changes were also arteriosclerosis related. Despite rapid fixation, examination of the renal tubular system was hampered due to autolytic changes.

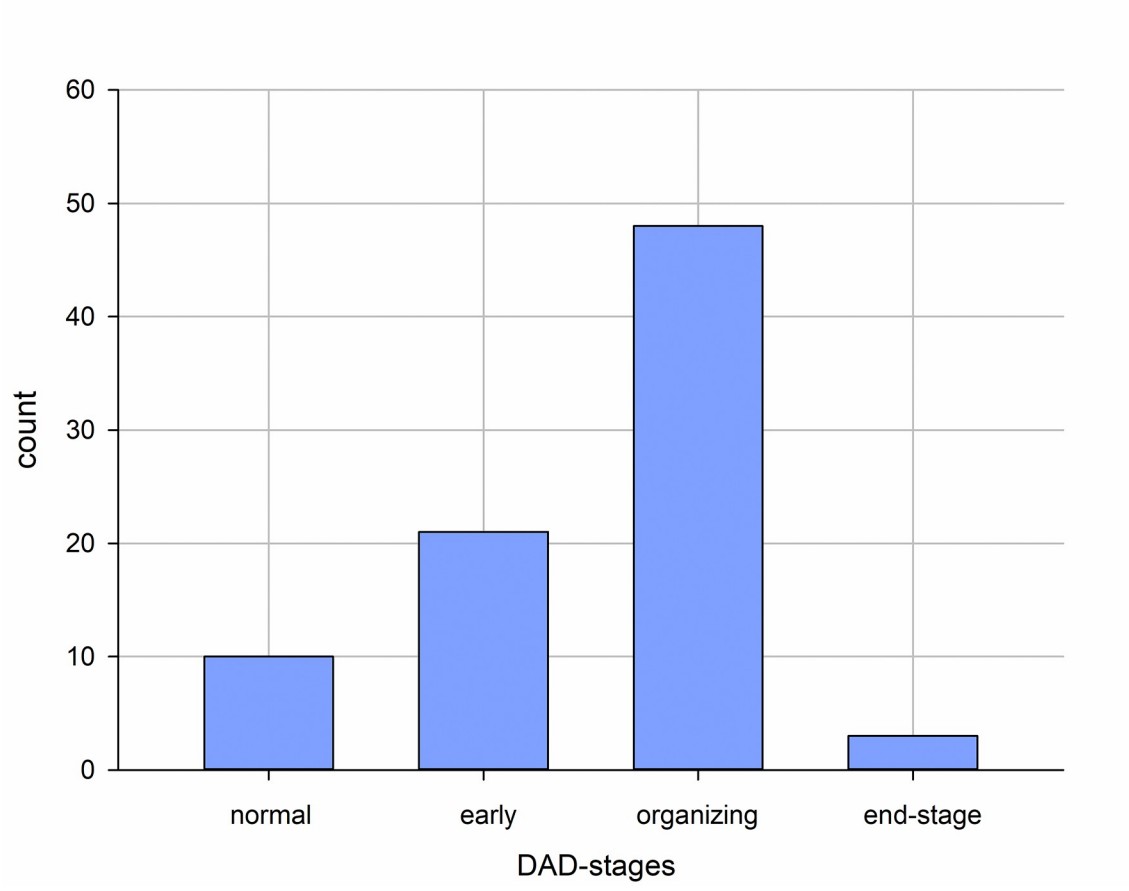

**Fig 1. Distribution of DAD based on 83 samples from 19 patients.** DAD = diffuse alveolar damage.

**CNS.** 225 samples of the autopsy of the brains, which was performed in 15 of the 19 cases showed no distinct changes of viral-mediated encephalitis or vasculitis within these samples from all parts of the brain and the spinal cord. The main findings were brain edema, hypertensive angiopathy, and arteriosclerosis. In one case, an old media infarction was found. Another case showed multiple fresh hemorrhages with hypoxic ganglion cell damage, and two areas of older infarctions–again, these events were not considered to have caused death.

**Causes of death.** According to WHO guidelines [17], 18 of 19 patients died due to COVID-19, and one died due to acute heart failure with underlying coronary artery disease, cardiomyopathy, and atrial fibrillation. Only very limited DAD was noted in the lungs (acute DAD in just one lobe in a limited area) in this case.

In 12 cases, death in the context of respiratory insufficiency with the presence of relevant DAD was to be assumed on the basis of the changes seen at autopsy and in consideration of the clinical data, in some cases combined with marked myocardial predamage. In the four partial autopsies, no definite cause of death was identified, but the combination of DAD and clinical data and course suggested death due to respiratory insufficiency. One patient died of hemorrhagic shock due to extensive hemopneumothorax with hemorrhage into the lungs on the floor of the extensive changes of the preceding COVID-19 pneumonia. Only one patient died due to thromboembolic events (mesenteric ischemia of the small intestine and fulminant

pulmonary artery embolism) in the context of the COVID-19 infection, accompanied by relevant signs of organizing DAD in three lobes.

**Viral mapping of the organs.** Mapping of the results of RT-qPCR, RNA-ISH, IHC, and histology of all organs, together with basic demographic data, are illustrated in Fig 2. Upon this basis, all further analyses were performed.

**Distribution of SARS-CoV-2 by RT-qPCR and correlation with histological changes.** In total, 362 samples (51 swabs, 32 body fluids, and 279 tissues) from 24 locations were tested for SARS-CoV2 with RT-qPCR to investigate the virus's spread and distribution within the organism. Altogether RT-qPCR revealed a positive result for SARS-CoV2 in 98 samples (32%).

At autopsy, the swabs were positive for SARS-CoV2-PCR in 15/19 cases (mean Ct-value and standard deviation (SD): 23.6 ± 5.9) from the pharynx, in 13/16 cases (mean Ct-value and SD: 25.5 ± 4.9) from the trachea, and in 11/16 cases (mean Ct-value and SD: 26.0 ± 5.8) from bronchus. The categorized Ct values of these swabs correlated strongly with those of the lungs (R > 0,82, p < 0.0001).

Pleural effusions were present in nine cases, five of which tested positive for SARS-CoV-2 by PCR (mean Ct-value and SD: 25.5 ± 4.9). Pericardial effusion was present in four cases, of which one was positive (Ct-value: 25.9). Peritoneal effusion was positive in one of three cases (Ct-value: 29.7). All 15 cerebrospinal fluid (CSF) samples were negative.

The distribution of SARS-CoV2 within the tissue samples evaluated using RT-qPCR is presented in Figs 2 and 3. The infection was mainly distributed in the upper airways and the lungs. This is true for both the number of positive samples and the viral load demonstrated by low Ct values. This was followed by intrathoracic locations (mediastinal lymph nodes and fat), which were affected by the virus in high frequencies ($\geq$ 60%) with low measured Ct values. The heart, coronary artery, central arteries liver, kidney, and spleen rarely exhibited a viral infection.

RT-qPCR-based analyses were performed on 60 samples from 15 brains with only one location revealing positivity at the bulbus olfactorius with a high Ct-value of 31. No histological changes were identified in this sample.

A significant inverse correlation was noted between stage of DAD (early, organizing, end-stage) and RT-qPCR Ct values on a case basis (R = -0.69, p = 0.001) and sample basis (R = -0.41, p < 0.001) (Fig 4A). On the level of cases and samples, significant correlations were also seen between interstitial lymphocyte infiltration and DAD (R = 0.47, p = 0.04 and R = 0.44, p < 0.001) but not with the RT-qPCR results. Moreover, Ct values did not correlate with perivascular lymphocyte infiltration (R = 0.09, p = 0.41), diffuse hemorrhage (R = - 0.128, p = 0.248), and microthrombi (R = -0.121, p = 0.274), nor did these histological factors correlate with each other (Fig 4B). In all other organs or regions, including trachea/ bronchi, liver, kidneys, spleen, mesenterial fat, and lymph nodes, large arteries, only the heart samples showed a highly significant correlation between mild inflammation in the epicardial region and PCR-based viral detection (R = 0.828, p < 0.001).

**Distribution of SARS-CoV-2 by RNA-ISH and IHC and correlation of the methods.** Besides RT-qPCR, SARS-CoV-2 infection of the organs was evaluated with in situ hybridization using RNA-ISH and IHC in representative samples from all lung lobes and in RT-qPCR-positive tissue samples outside of the lung.

In the lung sample, in addition to the positive-sense RNA, the negative-sense RNA was evaluated to identify the replicable virus by RNA-ISH. Representative images are shown in Fig 5. The analysis showed moderate to strong correlations between the four methods (R: 0.43–0.83, P: 0.06 to <0.001) with the highest positivity rate for RT-qPCR in 13 of 19 (68%) versus 10 (53%) positive cases in RNA-ISH positive-sense analysis and lowest rates in RNA-ISH negative-sense (3 of 19, 16%) and IHC (4 of 19, 21%) (Fig 4B). None of the six negative cases by

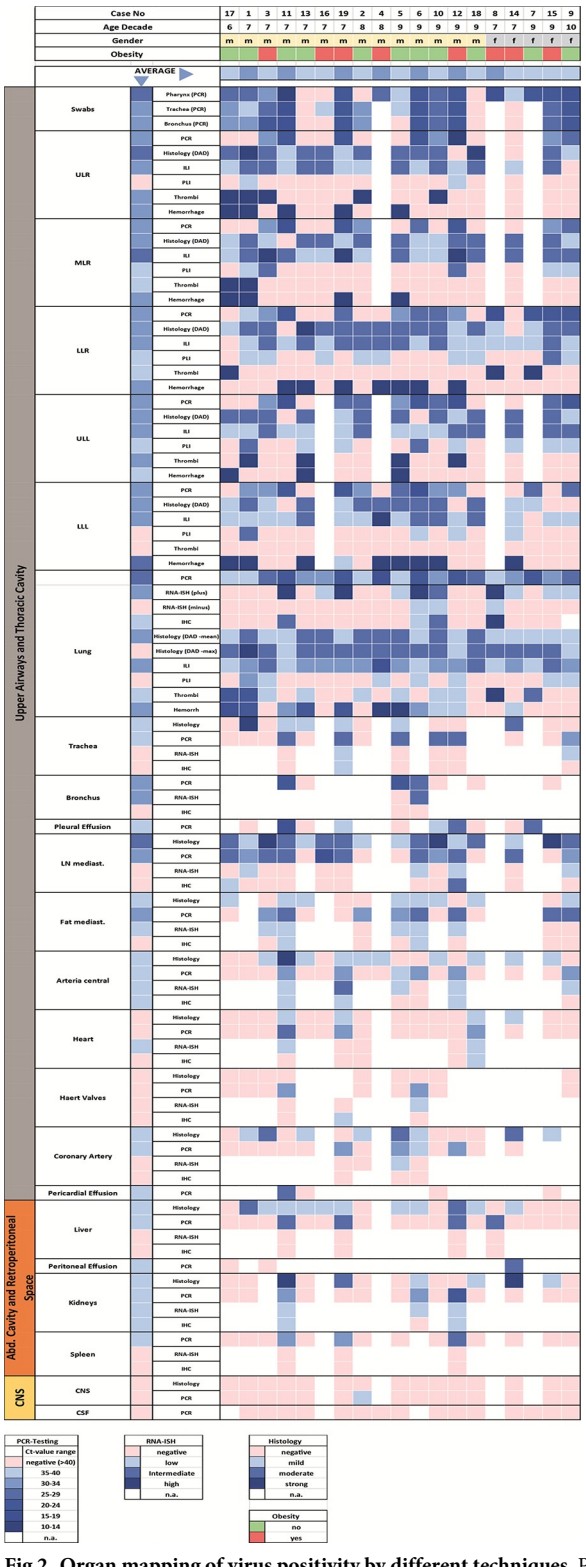

**Fig 2. Organ mapping of virus positivity by different techniques.** PCR = real—time quantitative polymerase chain reaction, RNA-ISH = RNA-in situ hybridization, IHC = immunohistochemistry, PLI = perivascular lymphocyte infiltration, ILI = interstitial lymphocyte infiltration, DAD = diffuse alveolar damage, ULL = upper left lobe, LLL = lower left lobe, LRL = lower right lobe, URL = upper right lobe, MRL = middle right lobe, CNS = central nervous system, CSF = cerebrospinal fluid); The category Histology represents lymphoplasmacellular inflammation of

the respective organ system, regardless of the compartment in a four tired quantification. Thrombi are classified as being present or absent.

RT-qPCR indicated a positive result using the other methods. To classify the viral affected cells in the lungs, IHC-double-staining have been performed. It could be shown that the main affected cells are pneumocytes with distinct nuclear expression of TTF-1 and perinuclear identification of viral protein (Fig 5). Next to that CD68-positive histocytes were also infected but at a lower frequency.

Sixty-one samples from other organs, confirmed positive by RT-qPCR, were parallel evaluated using RNA-ISH positive-sense and IHC with positivity rates of 49% (n = 30) and 28% (n = 17).

**Correlation of DAD with clinical parameters.** We further evaluated the correlations between DAD and clinical parameters. We found no correlation between DAD and age, smoking history, and BMI. Further, the number of comorbidities (p = 0.066) and the length of ventilation (p = 0.087) indicated a slightly not significant trend for a correlation with DAD. Sex (p = 0.036) and diabetes (p = 0.040) were the only variables that were correlated with DAD. DAD was 1.3 points higher in mean for men than for women, whereas diabetes reduced DAD by 1.2 points compared with nondiabetic patients.

We further fitted a multiple linear regression model with diabetes and sex which were both significant. Here, DAD was 1.5 points higher in men than in women (SE: 0.5, p = 0.004) and was decreased by 1.4 points for patients with diabetes (SE: 0.4, p = 0.005).

To determine the impact of these variables on DAD adjusted for other variables, we added the number of comorbidities and length of ventilation to the model. In this model sex and diabetes remained the only significant variables.

## Discussion

To the best of our knowledge, this is the first report of a COVID-19 autopsy series providing multiorgan SARS-CoV-2 mapping, including body fluids, with systematic histology, IHC,

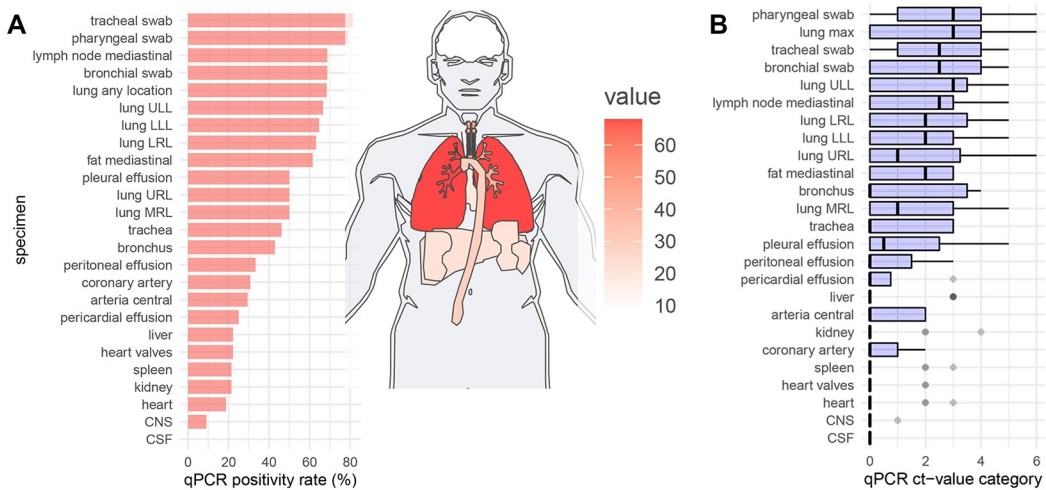

**Fig 3. Detection of SARS-CoV-2 in the different organs by RT-qPCR on a sample basis.** A) Positivity rata and distribution. B) Ct value categories. ULL = upper left lobe, LLL = lower left lobe, LRL = lower right lobe, URL = upper right lobe, MRL = middle right lobe, CNS = central nervous system, CSF = cerebrospinal fluid, qPCR = real-time quantitative polymerase chain reaction.

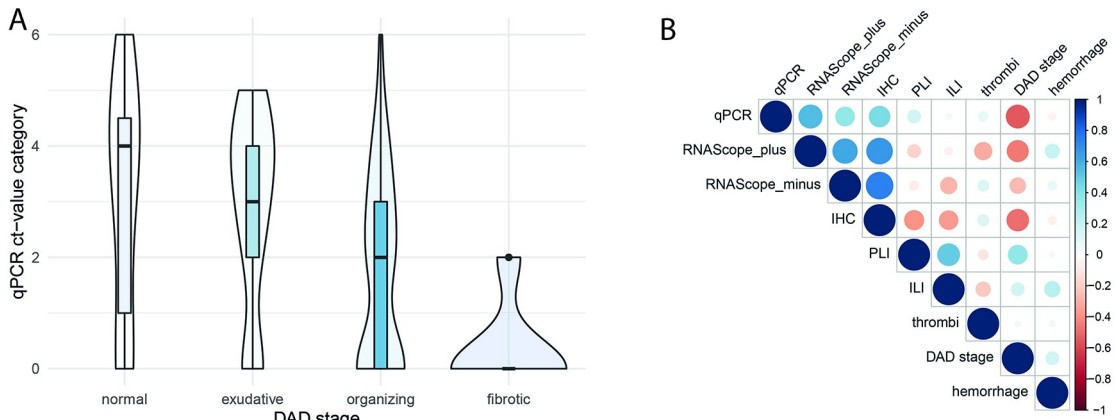

**Fig 4.** A) Correlation of DAD stage and RT-qPCR Ct values (Boxplot/Bean-plot) and B) Correlation of histological lung changes and RNA-detection with RT-qPCR and RNA-ISH and IHC on a sample basis. DAD = diffuse alveolar damage, qPCR = real-time quantitative polymerase chain reaction, ct = cycling threshold, IHC = immunhistochemistry, PLI = perivascular lymphocyte infiltration, ILI = interstitial lymphocyte infiltration.

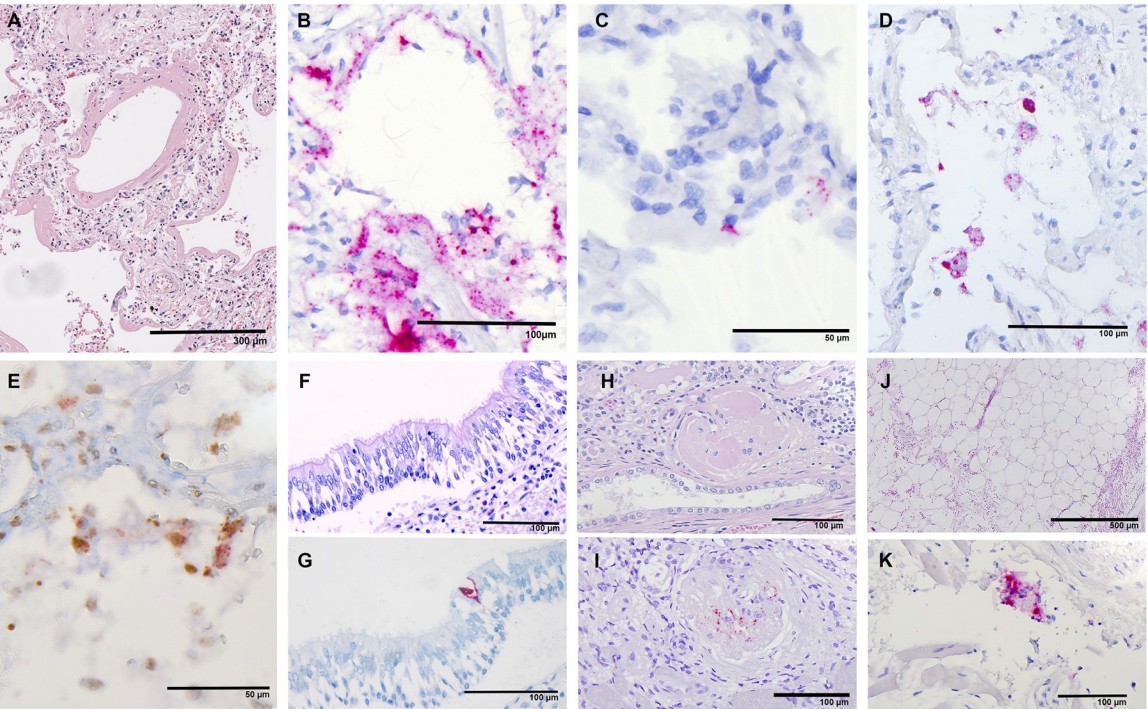

**Fig 5. Representative microscopic images.** A) H&E–Lung sample (case 8) with an early DAD with broad hyaline membranes, broadening of the alveolar septa, and mild lymphocytic inflammation. B) RNA-ISH positive-sense–same case as A); remarkable extensive SARS-CoV-2 detection in pneumocytes and interstitium. C) RNA-ISH negative-sense–the same case as A); negative -sense RNA is observed only in a few cells. F) H&E and G) RNA-ISH positive-sense; Conventional staining shows a normal mucosa with well-preserved respiratory epithelium. Detection of single infected cell with RNA-ISH. H) H&E and I) RNA-ISH positive-sense; Sclerosed gleomerulus with several infected cells. J) H&E and K) RNA-ISH positive-sense; Adipose tissue with mild hemorrhagic alteration but without relevant inflammation. High magnification of this slide reveals a single SARS-CoV-2 positive cell representation probably a histiocyte.

RT-qPCR, and RNA-ISH examinations [18]. This comprehensive approach to investigating many relevant organ systems within one study enables direct comparisons with a constant diagnostic setting that is not influenced by differing protocols or assays.

## Cohort

This study's collective represents 86% of all deceased with known SARS-CoV-2 infection in the time between April and May 2020 at University Medical Center Augsburg. One case came from a clinic outside Augsburg city and district. The pandemic situation was rather moderate at that time (7day incidence of max. 35/100,000) compared with other regions of Bavaria. The mean age in this cohort was lower (75 vs. 81 years) with a more male-predominant sex distribution (79% vs 53%) than the national data provided by the Robert Koch Institute [19]. This difference might be relevant because there are well-known sex-related differences of COVID [20, 21]. Our data also indicate that the women had a lower DAD stage than men. Other authors, however, could not identify any difference, at least for end-stage DAD [22]. Variants of concern [23] were not seen in Germany within the timeframe of this study. Sequencing of the virus was performed in five cases and revealed genome stability within the individual's respiratory tract (submitted for publication).

## Methods

In addition to H&E histology, we analyzed the tissue samples for SARS-CoV-2 by RT-qPCR, RNA-ISH, and IHC. In lung samples, in addition to the positive-sense RNA, the negative-sense RNA was targeted, indicating replication activity of SARS-CoV-2. In concordance with the literature, conventional histology did not reveal any virus-specific changes as noted in other viral infections, such as cytomegalovirus or herpes [18, 24]. RT-qPCR revealed viral RNA at a high frequency and can be assumed as very sensitive. The subsequently performed analyses using RNA-ISH and IHC identified viral RNA or protein with a specificity of 100%, but with substantially lower sensitivities in between 23% (IHC and RNA-IHC negative-sense) and 77% (RNA-ISH positive-sense) of the PCR-positive samples. Roden et al. recently reported a high sensitivity for droplet digital PCR and a considerably lower sensitivity by RNA-ISH and IHC of approximately one-third, which is comparable to our results [25]. On the basis of RT-qPCR as the gold standard, Massoth et al. compared RNA-ISH and IHC for the detection of this virus in 58 autoptic FFPE samples from different tissues. With sensitivity rates between 53% and 87%, their results are superior to ours [26]. These differences are likely due to technical differences in the protocols. First, we did not use automated platforms for RNA-ISH and IHC. Second, a different noncommercial antibody was used for IHC because of its very distinct staining results, resulting in reliable evaluability. In contrast to other reports, we identified no unspecific staining [18]. Because of the known difficulties of electron microscopy using autopsy material and the challenges of identifying the virus with certainty and the controversial debate regarding published electron microscopy results in the field of COVID-19, we decided not to initiate electron microscopy–related investigations [27, 28]. In concordance with other authors, we conclude that RT-qPCR is a highly sensitive and robust technique for screening for SARS-CoV-2 in tissues and liquid samples. It requires moderate technical efforts and is relatively cost effective. RNA-ISH is superior in terms of determining the exact localization of the virus and the tying of the affected cells, whereas IHC has lower cost and wider availability. The low sensitivity and low specificity of at least some antibodies hamper the benefits of this method.

## DAD

In this study, we noted severe changes in the lungs in all cases. Eighteen cases exhibited high grades of typical histological features of DAD. Only one case had relatively mild DAD changes, restricted to the left upper lobe. As mentioned earlier, this morphological change of COVID-19 has been confirmed in many reports [9–11, 24, 29–31]. DAD stages correlated inversely with the viral load (Fig 4). No correlations were identified for the extent of interstitial and perivascular lymphocytic infiltration or for the occurrence of microthrombi or diffuse hemorrhage. To the best of our knowledge, this is the only study to investigate such relations.

## Viral mapping

The viral infection within the body showed a strong accentuation within the upper airway and the lungs. Many authors have recently reported viral detection within the respiratory system [9, 10, 12, 24, 29–34]. Interestingly, a strong positive correlation of RT-qPCR between lung tissue and the swabs of throat, trachea, and bronchus was given. In only one of four negative pharyngeal swabs, the RT-qPCR testing of the lungs revealed a positive result. Despite the fact of viral spreading from the upper to lower airway, the virus or at least parts of the virus remain detectable in the upper airway at the time of death. Skok et al. reported almost identical results [34] relativizing the opposing results of some case and small series reports in this regard [35–37]. Our cohort's lower airways showed less often a positive ISH signal for negative-sense RNA in correlation to positive-sense RNA (3 vs. 13, 23%), indicating that not all detectable virus by RT-qPCR remains active. Massoth et al. reported a rate of two of 5 (40%) with positivity for the positive and negative-sense RNA. Interestingly, they reported negative-sense RNA by total RNA-seq in four of five cases, indicating a much higher sensitivity of sequencing.

Interestingly, mediastinal fat was infected at a high frequency with low Ct values. RNA-ISH revealed positive and negative-sense RNA that makes contamination unlikely. On the other hand, adipose tissue has a high density of the ACE-2-receptor [38, 39]. Together with spatial proximity, this may explain the unexpected high viral load. In the relatively rare cases of virus identification in heart samples, it was always found in the epicardial fat, not the myocardium. This agrees with other reports speculating epicardial fat's proinflammatory role in COVID-19 [40, 41]. Mediastinal lymph nodes showed viral infection at a high frequency, with RNA identification in almost 70% by RT-qPCR and 25% by RNA-ISH. An identical positivity rate by RT-qPCR was reported by Haslbauer et al. [42]. Other organs of the thoracal cavity were affected at a lower frequency and with a lower viral load. The evaluation of the heart (endo-, myo- and epicardium) revealed viral RNA in 3 of 16 samples (19%) by RT-qPCR and in 2 of those 3 also by RNA-ISH. These results are in between the findings of Bois et al., with no viral detection by droplet PCR (n = 15) [43] and the meta-analysis of Roshdy et al., with 47% (50 of 105) [44]. We paid special attention to heart valves and coronary arteries. The valves showed viral involvement at a similar frequency as the heart samples. Coronaries were involved in 4 of 13 (31%) samples. The three sample locations of the cardiac system did not correlate with each other regarding viral involvement.

Compared with the thoracal cavity, the involvement of the organs and structures of the peritoneal cavity and retroperitoneal space was considerably lower. The liver samples in our series were positive for SARS-CoV-2 in 22% (4 of 18) by RT-qPCR, whereas RNA-ISH and IHC revealed no positive case. Considerably higher positivity rates were reported by other authors, between 55% and 88% [24, 31, 45]. Given the high density of ACE receptors in the renal parenchyma, involvement by SARS-CoV2 was of particular interest at the beginning of the pandemic [33]. Nevertheless, whether direct infection or indirect effects are the main cause of renal dysfunction remains unclear [46]. The viral positivity rate in our series was 21% (3 of

14) by both RT-qPCR and RNA-ISH. In the largest series published so far, Braun et al. presented a higher rate of 60% (38 of 63 patients) [47]. The spleen, as another large lymphatic organ, shows a significantly lower viral load in the deceased compared with mediastinal lymph nodes. In only 3 of 14 spleens, viral RNA was identified by RT-qPCR with high to intermediate Ct values. Other authors have reported in series between 3 and 16 cases rates of splenic viral detection, ranging from 10% to 100% [24, 30, 31, 48]. The comparison with the mediastinal lymph nodes indicates that the detection of SARS-CoV-2 does not depend on the tissue type but on direct lymphatic drainage.

PCR analysis of brain tissue was performed in 15 cases. In only one case, PCR was positive, with a very high Ct-value. The rates in other published studies so far vary from our experience. Serrano et al. showed a frequency of viral detection in 20% of cases [49]. Analyzing the results of 28 autopsy studies, Li et al. reported a mean frequency RNA-based virus detection of 33% in 93 cases [8]. The rate of positivity, regardless of the method, was calculated to be 52%. The rate of viral detection in the cerebrospinal fluids of 468 patients was 6.4% [8]. We also analyzed CSF in 15 cases, and all of them were negative by RT-qPCR, which matches with the results of Serrano (9 analyzed CSF, all negative) [49]. Altogether, the published data are not consistent. The discussion of whether there is a direct brain involvement of the virus or not is open, but our data support the hypothesis of no direct involvement. This is also supported by a systematic review of Li [8].

## Detection of SARS-CoV-2 and correlation with histological changes

The only organ that showed a correlation between viral detection and morphological changes was the heart. Notably, this was restricted to the epicardium, which showed mild lymphocytic inflammation. This may have been caused by the high density of ACE-2 receptors in adipose tissue [38, 39]. The other histological changes, especially of the myocardium, coronary arteries, and valves, were correlates of preexisting conditions. Roshdy et al., including 316 cases over 41 studies, concluded that 1.5% of cases have myocarditis [44].

As reported above, viral RNA was detectable in other organs in varying frequencies. The consistently mild histological changes, however, did not correlate with RT-qPCR or RNA-ISH results but again represented preexisting conditions or could be understood as general changes due to a critical illness. Our results validate this, especially for the kidneys, liver, and brain. The kidney has been identified as a potential organ directly attached to the virus very early [10, 33]. Acute tubular damage is the most frequent histological change [6]. This was also noted in our series; however, based on our experience, this was found in similar frequencies in non-COVID autopsies. Furthermore, autopsy data alone may not be appropriate to derive this conclusion. The renal pathophysiology in COVID-19 requires further clarification, as also stated by other authors [46, 50]. The liver, an organ known for viral tropism and some associated characteristic histopathological patterns (e.g., CMV, EBV, and hepatitis), in this study cohort did not show any specific histological changes. Changes including mild portal chronic inflammation and steatosis can again be assigned as preexisting or acquired due to critical illness with multidrug therapy. This was also shown by Schmit et al. [51] in which they described only changes common for critically ill patients, drugs, or known preexisting diseases. By contrast, in a meta-analysis of 116 patients, not only steatosis (55.1%) but also congestion of hepatic sinuses (34.7%) and vascular thrombosis (29.4%) were noted [52]. Of the 19 cases, brains of 15 cases were examined histologically. In all cases, no specific abnormalities, especially in terms of encephalitis, was noted. This was also true in the olfactory bulb sample, which was revealed to be positive by RT-qPCR. Only one case showed several atypical hemorrhages. It seems not possible to discriminate whether this was caused by COVID-19 or coincidental. A vasculitis or

thromboembolic event was not found. Meinhardt et al. also identified viral RNA in approximately 10% of cases. Microthrombi and infarctions were detected in 18% of cases. Rather infrequent changes were also reported by Solomon et al. (inflammation in 4 of 18 cases) and Remmelink et al. (no inflammation in 11 cases) [24, 53]. Summarizing the results of several reports up to June and November 2020, Satturwar et al. and Mukerji et al. identified focal hemorrhage and hypoxic changes as main findings without evidence for virus-specific inflammation [6, 54]. The very recently published systematic review by Li et al. reports the results of 28 studies with abnormalities in 134 of 202 cases. Next to hypoxic changes and vascular lesions, a higher frequency of microgliosis/lymphatic infiltration was reported [8].

## Correlation of DAD with clinical parameters

Testing with different models, diabetes was associated with a lower DAD stage, whereas male patients presented a higher DAD stage than women. Other clinical parameters, such as age, smoking, BMI, number of comorbidities, and length of ventilation, had no significant effect on DAD. Diabetes in our cohort was present in 32% of the patients, which is in line with a large cohort of 5700 COVID-19 patients from the USA with 34% [55]. This comorbidity is associated with a higher mortality rate [56, 57]. This fact was also proven in another meta-analysis performed by Huang and colleagues [53], which showed that diabetes is associated with severe COVID-19 and developing ARDS. In this meta-analysis, which included 30 studies, sex did not affect the outcome, which is in contrast to our results. Altogether diabetes seems to be a strong risk factor for a severe course of COVID-19 and a bad outcome. This could be an explanation for the lower DAD in our cohort. Maybe patients with diabetes have a shorter time from diagnosis to death than nondiabetic patients. A time dependency of DAD patterns itself was published recently [58].

## Limitations

The remarkable high autopsy rate of 86% and the multimethodological approach are strengths of this study. However, there are also some limitations to mention. Although the results are relatively consistent within this cohort, conclusions should be drawn only with caution due to the small number of 19 cases from a single center. Long formalin fixing times and autolytic changes hamper the preanalytic conditions, which makes it difficult to estimate their influence on the results of IHC- and RNA-based analytics (to perform virus cultures was not possible for technical limitations). Finally, this study does not include all organs and tissues that might be of interest. This is especially true for the gastrointestinal tract. In our experience, these tissue types are subject to pronounced autolytic changes very early after death and were therefore excluded from further evaluations.

Another limitation concerning the comparison between the different techniques of viral detection (RT-qPCR, RNA-ISH and ICH) is the fact the organs beside the lungs only RT-qPCR positive samples were tested (due to the high number of samples) which could be a selection bias. But in the lungs at least no false negative RT-qPCR results were uncovered. But false positive RT-qPCR results could influence the calculated correlation between techniques.

## Conclusion

In this study, the lungs prove to be by far the organ system most affected by COVID-19. This is manifested at the histological level but also in the results of the various virus detection methods. Moreover, the representation of the negative sense RNA of SARS-Cov-2 here shows that replicable virus was still present even at the time of autopsy. More so, it was the failure of the respiratory system that led to the death of the patients. The conspicuous viral involvement of

the mediastinal adipose tissue may support the hypothesis of a role of fat as a proinflammatory factor postulated by other authors. In comparison, the other organ systems take a back seat. Here, only minor to moderate changes are found, most of which are not a direct consequence of viral damage with sufficient certainty. An indirect causality by a generalized immunological driven event seems more likely here. This could also involve the endothelial system. Because of the frequent reports of neurological symptoms in COVID-19 patients, we directed special attention to the CNS, but could not detect virus-induced changes here either and detected RNA with high Ct levels only in one case. Other methods are needed for the analysis of systemic effects of COVID-19 reported by clinicians, which may be mainly at the immunological and biochemical levels. We expect further insightful findings from currently ongoing mass spectrometry studies in the same collective.

## Supporting information

**S1 File. Protocol IHC.**
(PDF)

**S2 File. Protocol RNA-ISH.**
(PDF)

**S3 File. Protocol RT-qPCR.**
(PDF)

## Acknowledgments

We would like to thank for the technical support by Alexandra Martin, Elfriede Schwarz, Jenny Müller, Nadine Eismann, Juliane Torpier and Christian Beul of the Institute of General Pathology and Molecular Diagnostics, Medical Faculty, University of Augsburg, Augsburg, Germany. Further we thank Angele Breithaupt and Sven Reiche for the monoclonal SARS N-specific antibody and the protocol, Institute of Virology, University of Leipzig, Leipzig, Germany. The authors are particularly thankful to all relatives who gave their consent for the postmortem examination and thus made an invaluable contribution to the research of this new disease.

## Author Contributions

**Conceptualization:** Rainer Claus, Bruno Märkl, Tina Schaller.

**Formal analysis:** Klaus Hirschbühl, Stefan Schiele, Gernot Müller, Bruno Märkl.

**Funding acquisition:** Rainer Claus, Bruno Märkl, Tina Schaller.

**Investigation:** Klaus Hirschbühl, Sebastian Dintner, Claudia Wylezich, Jürgen Schlegel, Claire Delbridge, Lukas Borcherding, Jirina Lippert, Dimitra Moiraki, Oliver Spring, Michael Wittmann, Elisabeth Kling, Georg Braun, Thomas Kröncke, Rainer Claus, Bruno Märkl, Tina Schaller.

**Methodology:** Sebastian Dintner, Jürgen Schlegel, Claire Delbridge, Stefan Schiele, Gernot Müller, Tina Schaller.

**Project administration:** Bruno Märkl, Tina Schaller.

**Supervision:** Martin Beer.

**Writing – original draft:** Klaus Hirschbühl, Sebastian Dintner, Stefan Schiele, Bruno Märkl.

**Writing – review & editing:** Martin Beer, Claudia Wylezich, Rainer Claus, Tina Schaller.

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
