## [Decision Letter · Decision Letter 0]

23 May 2021

PONE-D-21-13695

Viral mapping in COVID-19 deceased in the Augsburg autopsy series of the first wave: a multiorgan and multimethodological approach

PLOS ONE

Dear Dr. Hirschbühl,

Thank you for submitting your manuscript to PLOS ONE. After careful consideration, we feel that it has merit but does not fully meet PLOS ONE’s publication criteria as it currently stands. Therefore, we invite you to submit a revised version of the manuscript that addresses the points raised during the review process.

Please pay special attention to the methods.

We look forward to receiving your revised manuscript.

Kind regards,

Etsuro Ito

Academic Editor

PLOS ONE

Journal Requirements:

2. Thank you for including your ethics statement:  "This study was approved by the internal review board of the Universitätsklinikum Augsburg (BKF No. 2020-18) and the ethics committee of the University of Munich (Project number 20-426, COVID-19 registry of the Universitätsklinikum Augsburg).".  

Please provide additional details regarding participant consent. In the ethics statement in the Methods and online submission information, please ensure that you have specified what type you obtained (for instance, written or verbal, and if verbal, how it was documented and witnessed). If the need for consent was waived by the ethics committee, please include this information.

Reviewers' comments:

Reviewer's Responses to Questions

**Comments to the Author**

1. Is the manuscript technically sound, and do the data support the conclusions?

Reviewer #1: Partly

Reviewer #2: Yes

2. Has the statistical analysis been performed appropriately and rigorously? 

Reviewer #1: Yes

Reviewer #2: Yes

3. Have the authors made all data underlying the findings in their manuscript fully available?

Reviewer #1: Yes

Reviewer #2: No

4. Is the manuscript presented in an intelligible fashion and written in standard English?

Reviewer #1: Yes

Reviewer #2: Yes

5. Review Comments to the Author

Reviewer #1: This series of 19 autopsies details the clinical characteristics, histologic findings, and distribution of SARS-CoV-2 in all organs by various testing modalities in patients who die of COVID-19. While the literature is ever expanding with descriptions of the pulmonary findings in these patients, I know of no other study that provides such a comprehensive look at viral load related to the histologic changes and correlated with the clinical parameters. I have a few comments below that if addressed, in my opinion, would add clarity to this work:

- The statement in the Introduction indicating that the clinical course of COVID-19 differs from other viruses (lines 77-78) should be elaborated upon as to "How?" and supported with references.

- In the Introduction, I think that the authors' aims could be more explicitly stated. Background is given, what they did is provided, but I seem to be missing the "Why?"

- Including Table 1 in the Materials and Methods Histology and IHC subsection (line 168) does not make sense to me, since this table is related to patient characteristics. This should be moved to a more appropriate subheading that describes the cohort.

- A table summarizing the lung and other organ histologic findings would be helpful. While the authors describe the scoring system used in the evaluation of these tissues in their materials and methods (lines 167-168), I see do not see where this is reported in their results section.

- The Discussion summarizes the findings, but I feel that 1) why this matters and 2) declarative statements about what the major conclusions that may be drawn from these are not addressed. To me, highlighting these would add significant value to the work rather than the discussion repeating the results section.

Reviewer #2: I would like to congratulate the authors with this valuable and excellent research. They investigated and characterised post-mortem pathological changes of major organs from patients deceased of COVID-19 and correlated this with viral presence in these organs. In addition, they validated and compared different techniques for viral detection. The authors presented the methods and data thoroughly. However, some questions and remarks remain.

Line 148 and Line 170 (methods) and Line 384 (Results): All lung samples were tested with all techniques, but only RT-PCR positive samples of other organs were tested with other techniques. I can understand that this was done out of practicality because of high number of samples and it would probably not change results regarding viral detection and histological changes. However, would this not induce some selection bias? Would you expect your correlation between techniques to be higher false-positive are not detected? Maybe this could be discussed further, or added to limitations?

Line 99 and Line 410 state that DAD is correlated to ‘some’ clinical parameters. I find the term ‘some’ rather vague. How many and which parameters were tested, and how were these parameters selected?

Line 120: In 4 patients, only a limited biopsy sampling has been done. Why was this done for these 4 patients?

Several test results have been categorized. How have these categories been chosen?

Line 170: How were representative samples chosen? Was this based on a set of criteria, or based on experienced judgement? Please clarify in the text.

Line 236: It can be useful to add information on treatment restrictions of patients, since there could be an impact of different treatment (or no treatment) on histological and viral changes?

Line 292: What was the reason for not assessing CNS in 4 patients? Please clarify in the text.

Line 293: “Many” samples. This is also rather vague. How many samples were evaluated, all?

Line 334: Virus detection was ‘successful’. Was it successful, and did other viral detection failed? Or was it positive and others negative?

I also think that another limitation would be that no viable virus detection was done. Of course, detection of viral RNA does not always relate to viable virus, and this could be an explanation why there were no correlations between histological changes.

Line 96: Organs instead of organisms?

Line 168: referral to evaluations in Table 1, but Table 1 are baseline patient characteristics.

Table 1 and 2: It is maybe worth considering to add percentages.

Line 289 ends with autolytic, I think there is a word missing.

Figure 2: “Haert” should be Heart. I would also suggest to add colour coding the other categories in the legends, as it will be easier for readers not to look for those in the text.

Line 335: “Swaps” should be swabs.

Line 469-470: This sentence misses a second part?

Figure 3: In the title it states ‘CT-values’ are added, but in the figure it states ‘CT-value categories’.

6. PLOS authors have the option to publish the peer review history of their article (what does this mean?). If published, this will include your full peer review and any attached files.

Reviewer #1: No

Reviewer #2: No

---

## [Author Response · Author response to Decision Letter 0]

17 Jun 2021

Rebuttal Letter – Response to Reviewers

Dear Professor Ito, 

many thanks to you and the reviewers for the substantial comments and suggestions on our manuscript, which were of great help to improve the conciseness and quality of the manuscript.

Concerning ethics statement and informed consent: 

Thank you for including your ethics statement: "This study was approved by the internal review board of the Universitätsklinikum Augsburg (BKF No. 2020-18) and the ethics committee of the University of Munich (Project number 20-426, COVID-19 registry of the Universitätsklinikum Augsburg).". 

Please provide additional details regarding participant consent. In the ethics statement in the Methods and online submission information, please ensure that you have specified what type you obtained (for instance, written or verbal, and if verbal, how it was documented and witnessed). If the need for consent was waived by the ethics committee, please include this information.

By law, a corpse is considered an object in Germany. According to national rules after the death of the patient, informed consent for the autopsy including research issues was collected verbal by next of kin and documented in the patient´s record. Informed consent from the alive patient in advance is not necessary according to the national rules. We have specified this in the methods section. (see line 126-130).

Please find below our point-by-point response to each comment of the reviewers:

Reviewer #1: This series of 19 autopsies details the clinical characteristics, histologic findings, and distribution of SARS-CoV-2 in all organs by various testing modalities in patients who die of COVID-19. While the literature is ever expanding with descriptions of the pulmonary findings in these patients, I know of no other study that provides such a comprehensive look at viral load related to the histologic changes and correlated with the clinical parameters. I have a few comments below that if addressed, in my opinion, would add clarity to this work:

Thank you very much for the comments, please find our additions and corrections point by point, hoping to properly address your criticisms:

- The statement in the Introduction indicating that the clinical course of COVID-19 differs from other viruses (lines 77-78) should be elaborated upon as to "How?" and supported with references.

We specified and added as follows: “COVID-19 presents more frequent with severe illness accompanied with lung injury which results more often in fatal course and also SARS-CoV-2 seems to be more contagious than influenza”. (see line 79-81). 

- In the Introduction, I think that the authors' aims could be more explicitly stated. Background is given, what they did is provided, but I seem to be missing the "Why?"

We specified and added as follows: “And this is true for COVID-19 as well, a new disease still not yet understood in detail and lack of sufficient therapeutic approach. Therefore, the aim was to analyze the macroscopic, microscopic changes and the spread of the virus in the deceased patients for the first step and to generate tissue and liquid samples for further investigations in terms of mechanisms of the disease.” (see line 85-89) 

- Including Table 1 in the Materials and Methods Histology and IHC subsection (line 168) does not make sense to me, since this table is related to patient characteristics. This should be moved to a more appropriate subheading that describes the cohort.

This was made by accident. We deleted the reference to Table 1 at this point. The correct place is in line 253.

- A table summarizing the lung and other organ histologic findings would be helpful. While the authors describe the scoring system used in the evaluation of these tissues in their materials and methods (lines 167-168), I see do not see where this is reported in their results section.

Thank you very much for this comment. The histologic findings of all organs including the lung, categorized by the described scoring system, is presented in the heat map (Fig 2). Unfortunately, we didn´t provide a colour code yet for the histologic findings, so we added this to the heat map (see Fig 2). Hopefully, this is clear for the readers now.

- The Discussion summarizes the findings, but I feel that 1) why this matters and 2) declarative statements about what the major conclusions that may be drawn from these are not addressed. To me, highlighting these would add significant value to the work rather than the discussion repeating the results section.

Again, thank you for the comment. You are right, the results are partly repeated in the discussion, but also compared with the findings from the literature. The order of the topics is almost the same as in the results section because there are so many points and we wanted to give a good structure for the readers. In order to obtain a well-rounded conclusion from the data discussed, we have renewed the conclusion section with the intention of placing the main findings of our work in the overall context of COVID-19. (see line 647-662)

Reviewer #2: I would like to congratulate the authors with this valuable and excellent research. They investigated and characterised post-mortem pathological changes of major organs from patients deceased of COVID-19 and correlated this with viral presence in these organs. In addition, they validated and compared different techniques for viral detection. The authors presented the methods and data thoroughly. However, some questions and remarks remain.

Thank you very much for the comments, please find our additions and corrections point by point, hoping to properly address your criticisms:

Line 148 and Line 170 (methods) and Line 384 (Results): All lung samples were tested with all techniques, but only RT-PCR positive samples of other organs were tested with other techniques. I can understand that this was done out of practicality because of high number of samples and it would probably not change results regarding viral detection and histological changes. However, would this not induce some selection bias? Would you expect your correlation between techniques to be higher false-positive are not detected? Maybe this could be discussed further, or added to limitations?

Thank you very much for this comment. Yes, of course there is a selection bias. And the reason to perform analysis as described was indeed the high volume of samples. Looking to the lungs, we performed the other techniques in each sample. And neither IHC nor ISH was positive in a PCR-negative sample. So, the risk for missing false negative PCR results by not testing all samples with IHC/ISH seems to be low. But the problems with the false positive RT-qPCR results could not be excluded. We add a comment to the limitations. (see line 640-644)

Line 99 and Line 410 state that DAD is correlated to ‘some’ clinical parameters. I find the term ‘some’ rather vague. How many and which parameters were tested, and how were these parameters selected?

In line 99 (now 105) and 410 (now 435), we deleted “some”. In the methods section, we added a paragraph in which we describe the parameters and how those were chosen. (see line 216-221).

Line 120: In 4 patients, only a limited biopsy sampling has been done. Why was this done for these 4 patients? 

This was due to limited permission by next of kin. We added: “Due to restricted consent in four cases, only limited autopsy with removal of parts of the lung, liver, kidney, spleen was performed. In these cases also no CNS specimen could be collected”. (see line 128-130)

Several test results have been categorized. How have these categories been chosen?

For histology, IHC and ISH, after first screening of the slides and the obtained information on the sectra of the changes, the categories have been defined. For PCR, the categories were abitrary divided into increments of 5.

Line 170: How were representative samples chosen? Was this based on a set of criteria, or based on experienced judgement? Please clarify in the text.

The selection was done by experienced judgement. We specified as follows: “All slides were evaluated by an experienced board-certified pathologist (TS). Based on the information obtained from this evaluation, particularly the extent of change seen, representative slides were selected and independently analyzed and classified by TS and another experienced board-certified pathologist (BM).” (see line 168-172).

Line 236: It can be useful to add information on treatment restrictions of patients, since there could be an impact of different treatment (or no treatment) on histological and viral changes?

We added in the text: “In four cases, therapy was limited at the patient's request and intensive medical treatment was not initiated.” (see line 261-262)

Line 292: What was the reason for not assessing CNS in 4 patients? Please clarify in the text.

The reason was the limited autopsy without brain autopsy due to restricted permission by next of kin. We clarified this in the methods section as follows: “Due to restricted consent in four cases, only limited autopsy with removal of parts of the lung, liver, kidney, spleen was performed. In these cases also no CNS specimen could be collected.” (see line 128-130)

Line 293: “Many” samples. This is also rather vague. How many samples were evaluated, all?

All 225 were evaluated, so we corrected as follows: “225 samples of the autopsy of the brains, which was performed in 15 of the 19 cases showed no distinct changes of viral-mediated encephalitis or vasculitis within these samples from all parts of the brain and the spinal cord.” (see line 317-319)

Line 334: Virus detection was ‘successful’. Was it successful, and did other viral detection failed? Or was it positive and others negative?

Positive and negative, we corrected as follows: “Altogether RT-qPCR revealed a positive result for SARS CoV2 in 98 samples (32%).” (see line 358)

I also think that another limitation would be that no viable virus detection was done. Of course, detection of viral RNA does not always relate to viable virus, and this could be an explanation why there were no correlations between histological changes.

We were not able to perform virus cultures because this would need a special security laboratory which is not established in our institute, and also not in others of the University Hospital of Ausburg. Certainly, this is a limitation. Therefore, RNA-ISH including the minus strand was performed as an alternative. This is at least better than only PCR analysis. At least for the lungs it can be said that despite pronounced histological changes, based on the minus strand analysis presumably no vital virus was present anymore. And since there were hardly any histological changes in the other organs, we do not believe that vital virus particles were present. We added a comment in the limitations paragraph. (see line 635-636)

Line 96: Organs instead of organisms?

With organisms we mean the body, and organisms for us seems to be the best word in this context. So we would rather leave it as it is. (see line 102)

Line 168: referral to evaluations in Table 1, but Table 1 are baseline patient characteristics.

This was made by accident. We deleted the reference to Table 1 at this point. The correct place is in line 253)

Table 1 and 2: It is maybe worth considering to add percentages.

Thank you very much for the hint, we added the percentages in both tables.

Line 289 ends with autolytic, I think there is a word missing.

…. autolytic “changes” was added. (see line 314)

Figure 2: “Haert” should be Heart. I would also suggest to add colour coding the other categories in the legends, as it will be easier for readers not to look for those in the text.

Heart was corrected and the colour codings for the other categories were also added. (see Fig 2) 

Line 335: “Swaps” should be swabs.

Corrected to swabs (see line 359)

Line 469-470: This sentence misses a second part?

This sentence and the following are now linked together. (see line 495-498)

Figure 3: In the title it states ‘CT-values’ are added, but in the figure it states ‘CT-value categories’.

In the title it was corrected: Ct value categories (see Fig 3)

So, once again, thank you very much for the valuable comments. We hope we were able to address them well.

With kind regards,

Klaus Hirschbühl

Corresponding author: 

Dr. med. Klaus Hirschbühl

Hematology and Oncology

Medical Faculty, University of Augsburg 

Stenglinstraße 2 

86156 Augsburg, Germany 

Email: klaus.hirschbuehl@uk-augsburg.de

---

## [Decision Letter · Decision Letter 1]

6 Jul 2021

Viral mapping in COVID-19 deceased in the Augsburg autopsy series of the first wave: a multiorgan and multimethodological approach

PONE-D-21-13695R1

Dear Dr. Hirschbühl,

We’re pleased to inform you that your manuscript has been judged scientifically suitable for publication and will be formally accepted for publication once it meets all outstanding technical requirements.

Kind regards,

Etsuro Ito

Academic Editor

PLOS ONE

Reviewers' comments:

Reviewer's Responses to Questions

**Comments to the Author**

1. If the authors have adequately addressed your comments raised in a previous round of review and you feel that this manuscript is now acceptable for publication, you may indicate that here to bypass the “Comments to the Author” section, enter your conflict of interest statement in the “Confidential to Editor” section, and submit your "Accept" recommendation.

Reviewer #2: All comments have been addressed

2. Is the manuscript technically sound, and do the data support the conclusions?

Reviewer #2: Yes

3. Has the statistical analysis been performed appropriately and rigorously? 

Reviewer #2: Yes

4. Have the authors made all data underlying the findings in their manuscript fully available?

Reviewer #2: Yes

5. Is the manuscript presented in an intelligible fashion and written in standard English?

Reviewer #2: Yes

6. Review Comments to the Author

Reviewer #2: (No Response)

7. PLOS authors have the option to publish the peer review history of their article (what does this mean?). If published, this will include your full peer review and any attached files.

Reviewer #2: No

---

## [Editor Report · Acceptance letter]

9 Jul 2021

PONE-D-21-13695R1 

Viral mapping in COVID-19 deceased in the Augsburg autopsy series of the first wave: a multiorgan and multimethodological approach 

Dear Dr. Hirschbühl:

I'm pleased to inform you that your manuscript has been deemed suitable for publication in PLOS ONE. Congratulations! Your manuscript is now with our production department. 

Kind regards, 

on behalf of

Prof. Etsuro Ito 

Academic Editor

PLOS ONE